# Lysine Inhibits Hemolytic Activity of *Staphylococcus aureus* and Its Application in Food Model Contaminated with *Staphylococcus aureus*

**DOI:** 10.3390/toxins14120867

**Published:** 2022-12-09

**Authors:** Yangli Wan, Xiaowen Wang, Tianyi Bai, Xuting Zheng, Liu Yang, Qianhong Li, Xin Wang

**Affiliations:** College of Food Science and Engineering, Northwest A&F University, Xianyang 712100, China

**Keywords:** food safety, hemolytic activity, lysine, *Staphylococcus aureus*, α-hemolysin

## Abstract

**Highlights:**

**What are the main findings?**
Lysine inhibits hemolytic activity of *S. aureus*;Lysine inhibits the expression of toxin Hla of *S. aureus*;Lysine inhibits Hla oligomerization;Lysine attenuates *S. aureus* supernatant damage in mice and Caco-2 cells;Lysine inhibits the expression of Hla of *S. aureus* in a food model and had good biocompatibility.

**What is the implication of the main finding?**
Lysine has the potential to be used as an anti- *S. aureus* preparation in the food industry.

**Abstract:**

Alpha-hemolysin (Hla) is one of the important exotoxins of *Staphylococcus aureus* (*S. aureus)* and can be used as a target to reduce the virulence of *S. aureus*. This study explored the inhibitory effect of Lysine (Lys) on Hla and its application in food safety. Lys significantly inhibited the expression of Hla at sub-inhibitory concentrations and directly interacted with Hla to interfere with its oligomerization and thus significantly inhibited its hemolytic activity. Notably, Lys attenuated *S. aureus* damage to mouse small intestine and Caco-2 cells and delayed mouse mortality. In the food model, Lys inhibited the expression of Hla of *S. aureus* and had no significant effect on the sensory score. Moreover, Lys had no obvious damage effect on the main organs of mice, which indicated that Lys has good biocompatibility and has the potential to be used in the food industry as an anti-*S. aureus* preparation.

## 1. Introduction

*Staphylococcus aureus* is a common foodborne pathogen that produces a variety of toxins attached to food that can cause soft tissue infections and damage, which poses a significant threat to food safety and human health [1,2]. In addition, *S. aureus* is widely present in the environment, with even 30% of the healthy population carrying it [3]. When the opportunity arises, it causes various diseases such as endocarditis, pneumonia, keratitis and sepsis [4,5]. Notably, *S. aureus* occurs in up to 20% of the gut in healthy individuals and is detected at a significantly higher frequency in infant feces than in adults [6]. For the treatment of *S. aureus* infection, antibiotics are commonly used at present and this method has achieved certain curative effects. However, the disadvantages caused by the abuse of antibiotics has followed. For example, the frequency of methicillin-resistant *S. aureus* strains has increased and the efficacy of vancomycin treatment has weakened, while threatening the survival of beneficial flora in the gut [7,8]. This prompts us to find new strategies to treat the intestinal infection of *S. aureus*, an urgent problem that we need to solve.

Under normal circumstances, some diseases caused by *S. aureus* are caused by exotoxins, not complete bacteria. For example, *S. aureus* food poisoning is a gastrointestinal disease caused by human consumption of food contaminated with toxins produced by *S. aureus*. Even if *S. aureus* has been killed by conventional cooking, heat-resistant enterotoxins can still cause nausea, vomiting and stomach cramps or diarrhea in humans [9]. In addition to enterotoxins, chromosomally encoded α-hemolysin are the major virulence factors of *S. aureus*, which are secreted as water-soluble monomeric (33.2 kDa) proteins that bind to the ADAM10 receptor on the surface of target cells without further modification, followed by the formation of a heptameric (232.4 kDa) transmembrane pore [10], causing cell Lysis and apoptosis [11,12]. Studies have shown that Hla can promote sepsis and intestinal inflammation by disrupting the integrity of the intestinal epithelial barrier to translocate bacterial cells and their products [13]. This suggests that we can use the exotoxin of *S. aureus* as a potential antibacterial target.

Lys, with the chemical name 2,6-diaminoacetic acid, is an essential amino acid for humans. Lys has important nutritional functions involved in protein synthesis, energy metabolism and immunity enhancement [14]. Lys can be used in food processing as food flavoring agent, coloring agent and nutritional fortifier. The Chinese national standard GB2760-2014 shows that, when Lys is used as a natural flavor in the food industry, there is no limit requirement for it [15]. Studies have shown that continuous administration of L-Lys at a dose of 16.8–17.5 g/d for a period of 1–1095 days has no statistical significance in the incidence of gastrointestinal symptoms [14]. At present, Lys has been widely studied in the fields of improving meat quality, supporting nutritional breeding and food processing [15]. However, in terms of food safety, Lys is mostly used in the research of food anti-bacterials in the form of polymer (ε-poly-lysine) [16,17]. Therefore, the purpose of this study was to evaluate the antibacterial activity of mono-Lysine against *S. aureus* and to further explore the possible mechanisms, using Hla as the entry point.

## 2. Results

### 2.1. MIC and MBC

The MIC of Lys to the four tested strains of *S. aureus* (29213, 265, 265Δ*sea* and 265Δ*hla* strains) were all 400 mM (Table 1) and the MBC was also 400 mM. This is much greater than 1024 μg/mL, indicating that Lys lacks good inhibitory activity against *S. aureus*.

### 2.2. Growth Curves and SIC

As shown in Figure 1B, Lys completely inhibited the growth of strain 265 at MIC and 2 MIC concentrations. Lys prolonged bacterial lag phase at 1/2 MIC, 1/4 MIC and 1/8 MIC concentrations. Compared with the control group, Lys at concentrations of 1/16 MIC, 1/32 MIC, 1/64 MIC had no significant effect on the growth of the 265 strain. Therefore, 1/16 MIC, 1/32 MIC, 1/64 MIC were defined as the SIC of Lys against *S. aureus* 265 strain.

### 2.3. Lys Inhibits Hemolytic Activity of S. aureus

The experimental results of the effect of Lys on the hemolytic activity of *S. aureus* at the concentration without anti-*S. aureus* activity are shown in Figure 1C,D. Lys significantly inhibited the hemolytic activity of *S. aureus* and proved dose-dependent. It is worth noting that the hemolytic activity of *S. aureus* was almost completely inhibited after co-culture of 1/16 MIC concentration of Lys. The results of Coomassie brilliant blue staining showed that the intracellular protein of *S. aureus* decreased after Lys treatment (Figure 1E) and western blot results also showed that Lys inhibited the expression of Hla protein of *S. aureus* in a dose-dependent manner (Figure 1F,G). Then to further analyze the reason why Lys inhibited hemolysis, Lys was directly co-incubated with *S. aureus* supernatant. As expected, Lys also had a significant inhibitory effect on the hemolytic activity of *S. aureus* supernatant in a dose-dependent manner (Figure 2A,B), which may be the key factor in Lys inhibiting the hemolytic activity of *S. aureus*.

### 2.4. Lys Inhibits the Hemolytic Activity of Hla and Interferes with the Oligomerization of Hla

To determine whether Lys inhibited the hemolytic activity of *S. aureus* due to direct interaction with Hla, the effect of Lys on the hemolytic activity of purified Hla was analyzed. As shown in Figure 2C,D, 100 mM Lys reduced the hemolysis rate of purified Hla to 7.22% and even 25 mM Lys weakened the hemolysis rate (83.90%) of purified Hla, which proved that Lys can directly combine with Hla in the supernatant of *S. aureus* to inhibit its hemolytic activity. Subsequently, in order to analyze whether the inhibition of Lys on the hemolytic activity of *S. aureus* was related to the degree of oligomerization of Hla, deoxycholate-induced Hla oligomerization experiments were carried out. The results are shown in Figure 2E,F; different concentrations of Lys had an inhibitory effect on the oligomerization of Hla. When the concentration of Lys was 1/16 MIC, the inhibitory effect on oligomerization was the most obvious and the inhibitory effect gradually weakened with the decrease of the concentration. This result indicated that Lys could directly inhibit the hemolytic activity of *S. aureus* by interfering with the oligomerization of Hla.

### 2.5. Lys Attenuates the Damage of S. aureus Supernatant to Jejunum and Ileum and Delayed the Mortality of Mice

In order to analyze whether Lys can inhibit the hemolytic activity of *S. aureus* supernatant in vivo, animal experiments were carried out. As shown in Figure 3, the jejunum and ileum mucosa of mice in the control group (TSB group) were intact in structure, with conical villi, tightly arranged, neat striatum, orderly arrangement of epithelial absorptive cells and well-developed intestinal glands. However, the surface of the intestinal mucosa in the group that was given 265 strain supernatant was eroded, the striatum was irregular, the epithelial absorptive cells were arranged in a disorderly manner and a large number of villi fell off. Fortunately, this injury was relieved by concurrent gavage of Lys. In the presence of 1/16 MIC concentration of Lys, the 265 strain supernatant caused less damage to the mouse jejunum and ileum. With the decrease of Lys concentration, the effect of alleviating the injury of *S. aureus* supernatant in the jejunum and ileum of mice was weakened. Surprisingly, Lys not only alleviated the damage of *S. aureus* to the small intestine (jejunum and ileum) of mice (Figure 4), but also significantly delayed the mortality of mice (*p* < 0.01).

### 2.6. Lys Reduces the Cytotoxicity of S. aureus Supernatant and Promotes Caco-2 Cell Migration

Cytotoxicity experiments were then used to analyze whether Lys could protect Caco-2 cells from damage by *S. aureus* supernatants. As shown in Figure 5A, the supernatant of strain 265 had a strong inhibitory effect on cell viability after co-incubating with cells. However, in the presence of Lys, the inhibitory effect of *S. aureus* supernatant on Caco-2 cell viability was attenuated. Notably, treatment of Caco-2 cells with 1/32 MIC concentration of Lys alone (Figure 5B,C) significantly promoted cell migration compared with the control group (*p* < 0.001). Taken together, these results suggest that Lys may not only inhibit its damage to cells by directly interacting with Hla, but also by promoting cell migration.

### 2.7. Lys Inhibits the Expression of Hla of S. aureus in a Food Model

To further analyze the potential of Lys as an antibacterial agent in the food industry, Lys was smeared on the meat surface and inoculated with *S. aureus*. The results are shown in Figure 6B,C; Lys (1/32 MIC and 1/16 MIC) treatment significantly (*p* < 0.001) reduced the expression of Hla in meat in a dose-dependent manner. Meanwhile, Lys treatment had no significant effect on meat morphology (Figure 6A), which was also confirmed by sensory scoring (Table 2).

### 2.8. Lys Has Good Biocompatibility

The mice in the control group and the Lys experimental group had no disease or death after continuous gavage for 30 d and there was no significant difference in diet and other health conditions and behavioral activities. The results of H&E staining experiments are shown in Figure 6D. Compared with the control group, no cell damage was found in the heart, liver, spleen, lung and kidney organs of the mice in the Lys experimental group, the cell structure was clear and the shape of the nucleus was normal. The above results showed that Lys had good in vivo biocompatibility.

## 3. Discussion

*S. aureus* exists widely in nature and a little negligence will cause food contamination and a series of safety problems such as food poisoning. According to the US Centers for Disease Control and Prevention, the infection rate caused by *S. aureus* ranks second in the world and *Escherichia coli* ranks first. As one of the most concerned food-borne pathogens, *S. aureus* has also brought huge economic losses in China. For example, the udders of dairy cows are inflamed by *S. aureus*, resulting in the contamination of milk by *S. aureus*, which brings huge economic losses to farmers and threatens the safety of consumers [18]. The use of antibiotics is an effective measure against *S. aureus* infection, but the overuse of antibiotics has resulted in the continuous emergence of resistant strains [19], which necessitates the search for new antibacterial strategies.

At present, a large number of studies have used virulence factors as targets for anti-infection exploration, which can reduce the pressure on the survival of *S. aureus* and reduce the possibility of the emergence of drug-resistant strains [20,21]. Hemolysin is one of the important virulence factors of *S. aureus*. It can be divided into four types: α, β, γ and δ, according to different antigens. About 90% of the pathogenic *S. aureus* produce highly toxic bacterial exotoxin is α-hemolysin. The *S. aureus* 265 strain in this study expresses onlyα-hemolysin. The α-hemolysin can bind to the cell membranes of erythrocytes, leukocytes and endothelial cells and assemble to form heptamers. On the cell membrane, the heptamer selectively enters the hydrophobic region of the lipid bilayer, forming micropores that can facilitate the passage of ions and small molecules, allowing the ions contained in the cell to flow out, damaging the integrity of the cell membrane and then creating the cell lysis [22]. Studies have shown that *S. aureus* α-hemolysin impairs the intestinal epithelial barrier, thereby promoting translocation of luminal bacteria into the blood, which may exacerbate sepsis [13]. Our experimental results also confirmed the toxic effect of Hla (Figure 4). When the mice were injected intraperitoneally with the supernatant of *S. aureus* knocked out for the *hla* gene, no death occurred, in stark contrast to the supernatant of the wild-type strain 265 (100% death).

At present, drug research with α-hemolysin as the therapeutic target is divided into two strands: one is to reduce the expression of Hla; the other is to antagonize or neutralize the effect of Hla. In this study, Lys had no effect on the growth of *S. aureus* at the concentrations of 1/16 MIC, 1/32 MIC and 1/64 MIC, but significantly inhibited the hemolytic activity of *S. aureus*. The results of this study also indicate that Lys may inhibit the hemolytic activity of *S. aureus* by inhibiting the expression of Hla, or it may directly interact with Hla in the supernatant of *S. aureus* to inhibit the oligomerization of Hla and then inhibits its lysis effect on red blood cells [23,24]. This suggests that Lys is a potential anti-Hla drug. It should be noted that the specific binding sites of Hla and Lys need to be further studied in depth. Studies have shown that some antibiotics (such as β-lactam and fluoroquinolone antibiotics) can increase the expression of Hla at sub-inhibitory concentrations and the combination of these antibiotics with anti-Hla drugs can make up for the shortage of antibiotics [25]. Increased expression of Hla also increases the growth and reproduction of other bacteria (especially Gram-negative bacteria), aggravating the symptoms of *S. aureus* infection [26]. Therefore, the combination of antibiotics and anti-Hla drugs will be a hot topic in the future.

The intestinal tract is not only the main site of digestion and absorption, but also the main immune organ of the human body. It plays an important role in maintaining the normal immune function of the body and preventing the invasion of harmful substances [27]. In the present study, Lys treatment significantly alleviated the damage of *S. aureus* supernatant to mouse jejunum and ileum. This may reduce the damage to the intestinal barrier of mice and inhibit the translocation of *S. aureus* cells and other toxins, which may also be the reason why co-treatment of Lys with *S. aureus* delayed mouse mortality. In addition, combined with the staining results of Coomasil bright blue, Lys inhibited the expression of intracellular proteins, not only Hla, but also the expression of other toxins, which is another possible factor in Lys alleviating the damage of *S. aureus* supernatant to jejunum and ileum. Likewise, Lys also exhibited a high degree of protection against Caco-2 cells in cytotoxicity assays. Even more surprising is that Lys treatment alone not only has no toxic effect on cells, but even promotes the migration of Caco-2 cells. This result can also serve as a proof that Lys has a protective effect on the small intestine of mice.

As an essential amino acid and food additive, Lys has been extensively studied in supporting human health and improving food mechanisms. Although the poly-lysine formed by the polymerization of Lys monomer has high antibacterial activity, research on the antibacterial aspect of monomeric Lys is rare. In this study, the inhibitory effect of Lys monomer on the common food-borne pathogen-*S. aureus* was explored for the first time and it was found that its antibacterial ability was poor, but it was outstanding in inhibiting the hemolytic activity of *S. aureus*. In the application of food models, our results also showed that Lys pretreatment had no effect on the sensory scores of food models. At the same time, the results of acute toxicity experiments in mice also show that Lys has good biocompatibility, which is consistent with previous research results [14]. In conclusion, our study demonstrates the possibility of Lys as a novel strategy against *S. aureus* virulence in food, heralding the potential of Lys in playing a role in food safety.

## 4. Conclusions

As a virulence factor of *S. aureus*, Hla plays an important role in various infections of *S. aureus*. In this study, we evaluated the effect of Lys, a nutrient, on the hemolytic activity of *S. aureus* and explored the mechanisms involved. Our results indicated that Lys inhibited the hemolytic activity of *S. aureus* by inhibiting the expression of Hla and interfering with the oligomerization of Hla. This allows Lys to alleviate the damage of *S. aureus* supernatant to mice and Caco-2 cells. In food model applications, Lys exhibited the ability to inhibit Hla expression. At the same time, in animal toxicity experiments, Lys has no obvious toxic effect on the main tissues and organs of mice, indicating that Lys has good biocompatibility and has the potential to be used as an anti- *S. aureus* preparation in the food industry.

## 5. Materials and Methods

### 5.1. Strain and Culture

The strain sources are shown in Table 1. *S. aureus* strains were first inoculated on tryptic soy agar (TSA) plates and incubated at 37 °C for 24 h. A single colony was inoculated into 5 mL of tryptone soy broth (TSB) medium and continued to incubate for 16 h, so that the strain was in the logarithmic growth phase. The incubated cells were washed twice with sterile phosphate buffered saline (PBS, pH = 7.4), resuspended and diluted with TSB. The optical density (OD) value of the bacterial suspension was adjusted to 0.5 McFarland under the wavelength of 600 nm, that is, the bacterial concentration was 1 × 10^8^ colony forming unit (CFU) per mL. After the final bacterial suspension dilution concentration was adjusted to 1 × 10^6^ CFU/mL, which was used for subsequent experiments.

### 5.2. Minimum Inhibitory Concentration (MIC) and Minimum Bactericidal Concentration (MBC)

MIC was determined according to the microbroth dilution method with slight modifications [28]. Briefly, 100 μL of the prepared 1 × 10^6^ CFU/mL bacterial suspension was added to a 96-well plate, followed by different concentrations of Lys (Solarbio, SL8330), resulting in final Lys concentrations of 1600 mM, 800 mM, 400 mM, 200 mM, 100 mM, 50 mM, 25 mM, 12.5 mM, 0 mM. Wells containing only TSB medium were used as negative controls. The 96-well plate was incubated in a 37 °C incubator for 24 h and the OD value was measured at 600 nm (Multifunctional Enzyme marker, spark, Austria). The MIC was defined as the lowest Lys concentration at which no visible bacterial growth was detected in the medium. A 100 μL of culture in wells with no bacterial growth were spread on TSA plates and incubated at 37 °C for 24 h. MBC was defined as the lowest Lys concentration that reduces the total viable count to 99% or less of the initial viable count.

### 5.3. Growth Curve

Growth curves were determined according to the method described by previous research with slight modifications [29]. First, 265 bacterial suspensions with a concentration of 1 × 10^6^ CFU/mL were added with different concentrations of Lys, so that the final concentration of Lys was 2 MIC, MIC, 1/2 MIC, 1/4 MIC, 1/8 MIC, 1/16 MIC, 1/32 MIC, 1/64 MIC, 1/128 MIC and 0. After gentle mixing, different sample groups were added to 96-well plates and incubated at 37 °C for 24 h. At 600 nm, OD values were collected every 1 h and the final bacterial growth curve was plotted. The Lys concentration that had no effect on the growth of the *S. aureus* cells was defined as the sub-inhibitory concentration (SIC).

### 5.4. Hemo-ysis ana Lysis

The hemolytic activity experiment was based on the description of Jiang et al. with slight modifications [20]. Briefly, 265 bacterial suspensions with a concentration of 1 × 10^6^ CFU/mL were added with different concentrations of Lys, so that the final concentrations of Lys were 0, 1/16 MIC, 1/32 MIC, 1/ 64 MIC. The mixture was then incubated at 37 °C in a constant temperature shaker (180 rpm) for 24 h. After the cultures were centrifuged at 10,000 rpm for 10 min, the supernatant was collected. A 100 μL of supernatant was added to 900 μL of 6% sheep red blood cell solution (4 × 10^8^/mL). 0.1% Triton-X 100 (100% hemolyzed) was used as a positive control and TSB medium was used as a negative control. After incubating at 37 °C for 30 min, the mixture was centrifuged at 2500 rpm for 10 min and the supernatant was placed at 405 nm for OD determination.

For determination of hemolytic activity of *S. aureus* supernatants, a 10 mL of 265Δ*sea* bacterial suspension with a concentration of 1 × 10^6^ CFU/mL was added to 90 mL of brain heart infusion medium (BHI) and, after mixing, it was placed in a shaker at 37 °C for 36 h. The culture product was then centrifuged at 12,000 rpm for 20 min and the supernatant was filtered through a 0.22 μm filter and collected. Different doses of Lys were then added to 100 μL of the supernatant to achieve final Lys concentrations of 1600 mM, 800 mM, 400 mM, 200 mM, 100 mM and 0 mM. After the mixture was incubated at 37 °C for 30 min, 900 μL of 6 % sheep red blood cells was added and incubated at 37 °C for 30 min. The OD value at 405 nm was then determined as described above.

### 5.5. Coomassie Brilliant Blue Staining

The Coomassie brilliant blue staining method was performed as in previous research, with some modifications [30]. *S. aureus* was treated with sub-inhibitory concentration of Lys for 24 h, centrifuged at 12,000 rpm for 2 min and the cells were collected. The cells were then washed twice with PBS (pH = 7.4) and resuspended in PBS. Then the bacteria were placed in a cell sonication apparatus for disruption, the power was set to 35%, the ultrasonic was 4 s and the rest was 6 s for a total time of 10 min. Sonication was performed on ice throughout and protease inhibitors (PMSF) were added prior to sonication to prevent protein degradation. After sonication, the supernatant was collected by centrifugation at 12,000 rpm for 10 min at 4 °C. The protein concentration was determined according to the BCA method. 2 X protein loading buffer was added to the sample at a ratio of 1:1. After mixing, it was boiled in water at 100 °C for 5 min. A 10 μL of the sample was loaded into a protein gel (5% stacking gel, 12% resolving gel) and electrophoresed. The gel was stained in Coomassie brilliant blue for 1 h, followed by overnight de-staining, and finally the gel was imaged in a gel imager (GE1DOC XR+, Bio-Rad, Hercules, CA, USA).

### 5.6. Western Blot

Western blots were performed as described in previous research with slight modifications [31]. After *S. aureus* was treated with sub-inhibitory concentrations of Lys for 24 h, the supernatant was collected. Protein loading buffer was added to the supernatant and, after mixing, it was boiled in water at 100 °C for 5 min. Protein gel electrophoresis was then performed as described in 2.5. After electrophoresis, the proteins on the gel were transferred to PVDF membrane (0.22 μm, Millipore, MA, USA). PVDF membranes were incubated in Hla primary antibody (1:1000, Abcam, Cambridge, UK) overnight at 4 °C. Membranes were then washed 5 times with TBST and incubated in secondary antibody (1:5000, Abcam) for 1 h at room temperature. Membranes were washed again 6 times and imaged in a chemiluminescence imaging system (Champ Chemi 610 Plus, Beijing Saizhi Venture Technology Co, Beijing, China).

### 5.7. Oligomerization Experiment

To assess whether Lys could inhibit the ability of Hla to form heptamers, the oligomerization experiments followed previous research, with slight modifications [20]. Briefly, Lys was added to 5 mM deoxycholate to achieve final concentrations of 0, 1/16 MIC, 1/32 MIC and 1/64 MIC. Purified Hla was also added and the mixture was incubated at 22 °C for 20 min. A loading buffer without β-mercapto-ethanol was then added to the samples and the mixture was placed at 55 °C for 10 min. Next, western blot experiments were performed.

### 5.8. Intestinal Histology Score

The intestinal histological evaluation was conducted with reference to previous studies and combined with in vitro experimental data [32,33,34]. To evaluate whether Lys has protective effect on the intestinal tract of mice, different concentrations of Lys were added to the supernatant of strain 265 so that the final concentrations were 1/16 MIC, 1/32 MIC, 1/64 MIC and 0; the mixture was then gavaged to mice. After 24 h, the mice were euthanized by intraperitoneal injection of sodium pentobarbital (160 mg/kg) and the jejunum and ileum were collected and stained with H&E. Jejunal villus height, integrity and crypt depth were observed and recorded.

### 5.9. Mortality

The effect of Lys on the mortality of mice injected intraperitoneally with the supernatant of the strain was examined. Statistics of intraperitoneal mortality in mice were as described in previous research, with minor modifications [35]. Simply put, 33 healthy Kunming mice (6-weeks-old) were previously housed in a room for one week with a constant temperature of 23 °C and well-ventilated, during which the mice were allowed to eat and drink ad libitum. Mice were randomly divided into three groups: one group was injected with 200 μL of 265 strain supernatant, one group was injected with Lys (400 mM)-containing 265 supernatant and the other group was injected with 265Δ*hla* strain supernatant. Animals in each group were allowed to eat and drink freely and the number of deaths and the time of death were recorded regularly by professional experimenters. Finally, the cumulative mortality of mice in each group was counted.

### 5.10. Cytotoxicity Test

The method of Zhou et al. was used as a reference to determine the toxicity of Lys co-treatment with *S. aureus* supernatant on Caco-2 cells [36]. After Caco-2 cells were cultured to about 80% in complete medium (10% fetal bovine serum), they were dispersed in a 96-well plate at 1 × 10^4^ cells per well. After 12 h of culture, Caco-2 cells were treated with the supernatant of strain 265 containing different concentrations (1/16 MIC, 1/32 MIC, 1/64 MIC and 0) Lys for 24 h. It was discarded and replaced with fresh medium and then 10 μL of CCK8 reagent was added. After incubation at 37 °C for 2 h, the optical density was measured at 450 nm.

### 5.11. Scratch Test

The effect of Lys on the migration of Caco-2 cells was analyzed with reference to previous studies with appropriate modifications [37]. Caco-2 cells at a density of 1 × 10^5^/well were seeded into a 12-well plate. After overnight culture, a straight line was drawn with a 200 μL sterile pipette tip and the excess cells were washed away with PBS. Subsequently, the medium containing 1/64 MIC concentration of Lys was added and the culture was continued and photographed and observed at 0 and 24 h, respectively. The migration area was analyzed with ImageJ software (National Institutes of Health, Maryland, USA) and the mobility was calculated according to the following equation:Mobility(%) = (W_0_ − W)/W_0_ × 100.(1)
in which W_0_ is the average scratch area measured at 0 h and W is the average scratch area measured at 24 h.

### 5.12. Biocompatibility

The determination of Lys biocompatibility refers to previous studies with slight modifications [38]. Briefly, Kunming rats with the same health status by gender, weight and age were randomly divided into two groups, with 5 rats in each group. One group was gavaged with BHI medium alone and the other group was gavaged with BHI medium containing Lys (5 M/kg). After continuous gavage for 30 d, the mice were euthanized by intraperitoneal injection of sodium pentobarbital (160 mg/kg) and the heart, liver, spleen, lung and kidney were fixed and stained with H&E and finally observed under a light microscope (OLYMPUS, Tokyo, Japan).

### 5.13. Application of Lys in Food Model

Lamb was used as a food model to simulate the application of Lys in the food industry, as described in a previous study [39]. Lamb meat purchased from a local supermarket was cut into 2 × 3.5 cm pieces and then soaked in *S. aureus* suspensions (1 × 10^6^ CFU/mL) containing different concentrations of Lys (1/16 MIC, 1/32 MIC, 1/64 MIC, 0). The sample was taken out after 5 min. The soaked samples were first placed at 4 °C for 48 h and then placed at room temperature for 2 h. Finally, the samples were placed in PBS and ground to homogenize. After centrifugation at 12,000 rpm for 30 min at 4 °C, the supernatant was collected and the content of Hla protein in the samples was detected by western blotting.

The effect of Lys on sensory scores of food models was assessed as described by Ivanišová et al. with minor modifications [40]. Color, texture, taste and overall acceptability were used as the main indicators for sensory evaluation and a five-point scoring method was implemented. Twenty-one evaluators (10 males, 11 females) were professionally trained and scored and the sensory scoring criteria are shown in Appendix A.

### 5.14. Statistical Analysis

All experiments were repeated three times and data represent mean ± standard deviation. Significant data were analyzed by one-way ANOVA (*p* < 0.05) and Student’s *t*-test with SPSS software (USA). For two-group comparison, *p* values were derived from the one-way Student t test to determine differences between groups with normally distributed data. Data with normal distribution were analyzed by one-way ANOVA with Dunnett’s post-test or Tukey’s correction for multiple comparisons.

## Figures and Tables

**Figure 1 toxins-14-00867-f001:**
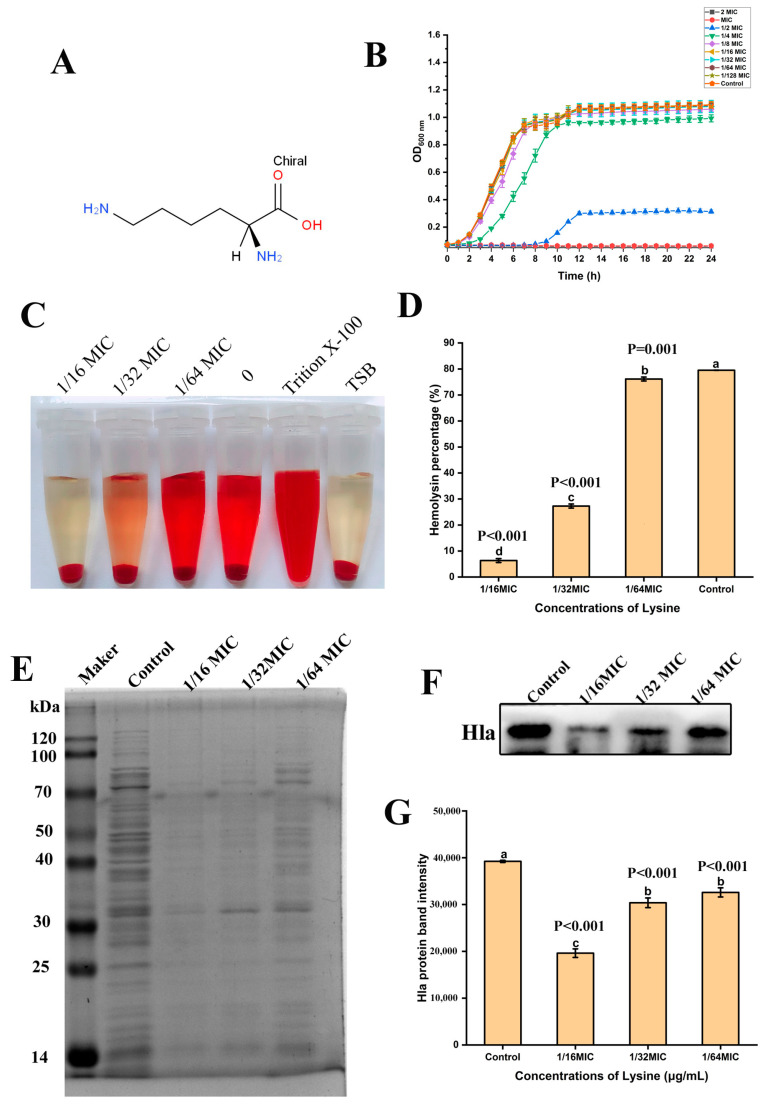
Lys inhibited hemolytic activity of *S. aureus*. (**A**) Molecular structural formula of Lys. (**B**) Growth curves of *S. aureus* 265 strains treated with different concentrations of Lys. There were significant differences between the mean values of unshared letters.(**C**,**D**) The effect of sub-inhibitory concentration of Lys on the hemolytic activity of *S. aureus* after 24 h treatment of *S. aureus*. (**E**) Coomassie brilliant blue staining to analyze the effect of sub-inhibitory concentration of Lys treatment on intracellular protein expression of *S. aureus*. (**F**,**G**) Western blots examined the effect of sub-inhibitory concentrations of Lys on Hla secretion from *S. aureus*. There were significant differences between the mean values of unshared letters.

**Figure 2 toxins-14-00867-f002:**
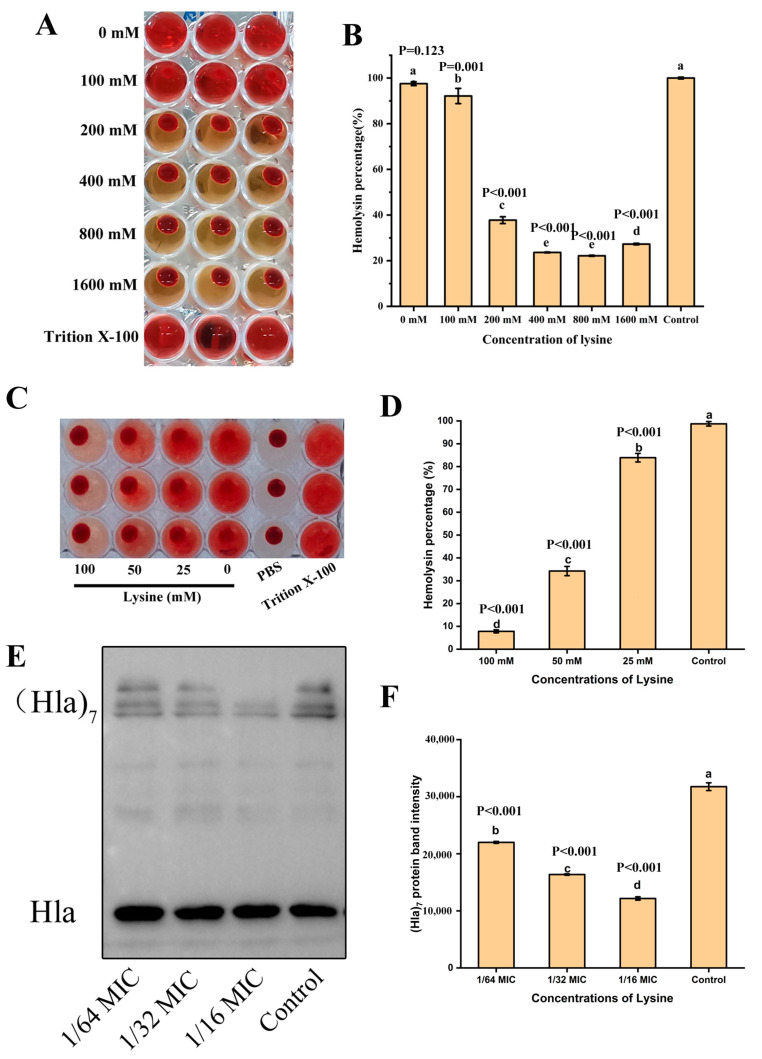
Lys directly interacts with Hla to inhibit hemolysis. (**A**) After co-incubation of different concentrations of Lys with *S. aureus* supernatants, hemolytic activity experiments were performed. (**B**) Statistical analysis of the effect of Lys on the hemolytic activity of *S. aureus* supernatants. (**C**) The effect of different concentrations of Lys on the hemolytic activity of Hla after co-incubation with purified Hla. There were significant differences between the mean values of unshared letters. (**D**) Statistical analysis of the effect of Lys on the hemolytic activity of purified Hla. After treatment of purified Hla with different Lys in the presence of deoxycholate. There were significant differences between the mean values of unshared letters. (**E**) Western blot was used to detect the oligomerization degree of Hla in different treatment groups. (**F**) Grayscale analysis of (Hla)_7_ protein band intensity. There were significant differences between the mean values of unshared letters. Note: 1600 mM is 4 MIC, 800 mM is 2 MIC, 400 mM is MIC, 200 mM is 1/2 MIC, 100 mM is 1/4 MIC.

**Figure 3 toxins-14-00867-f003:**
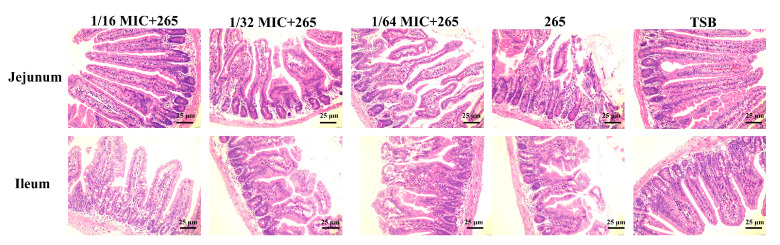
Lys alleviates *S. aureus* injury to jejunum and ileum in mice. Lys attenuates damage to mouse jejunum and ileum by *S. aureus* supernatant. The 265 strain supernatant was gavaged to mice in the presence or absence of Lys. After 6 h, mouse jejunum and ileum were collected and stained with H&E. Scale bar is 25 μm.

**Figure 4 toxins-14-00867-f004:**
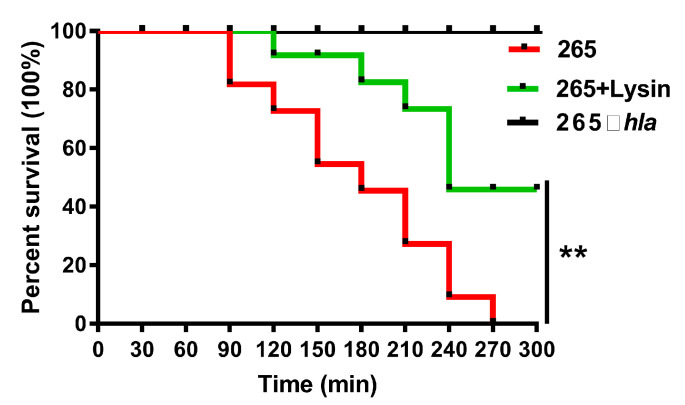
Lys delayed mouse mortality. The 265 strain supernatant was injected intraperitoneally into mice in the presence or absence of Lys. The number of dead mice was counted every 30 min and the survival rate of mice was summarized after 5 h. ** *p* < 0.01.

**Figure 5 toxins-14-00867-f005:**
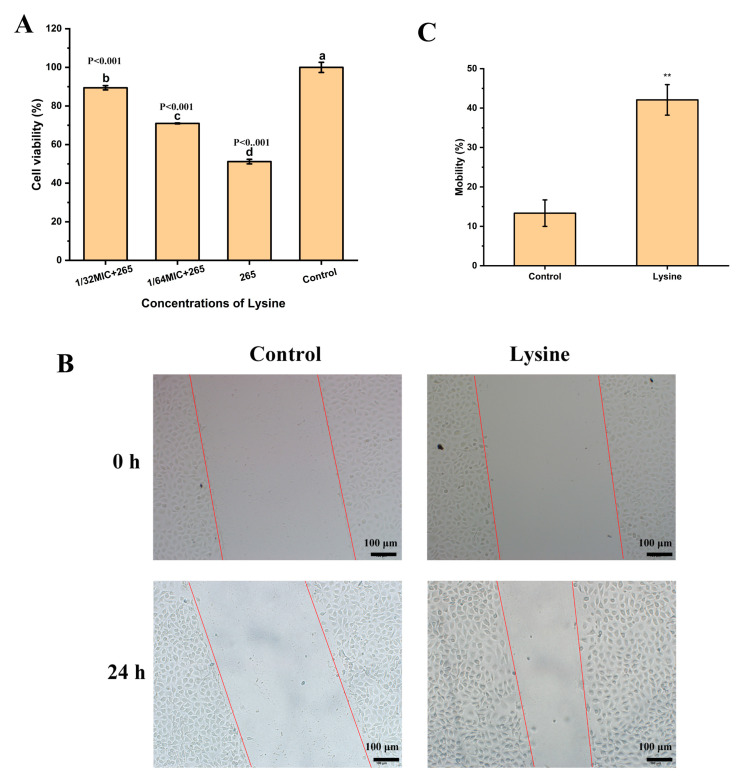
Lys alleviates *S. aureus* injury in Caco-2 cells. (**A**) Toxic effects of strain 265 supernatants on Caco-2 cells were examined in the presence or absence of Lys. There were significant differences between the mean values of unshared letters. (**B**) The effect of Lys treatment on Caco-2 cell migration was observed with an inverted microscope, magnification 200 X. (**C**) Cell mobility was analyzed with ImageJ software (National Institutes of Health, Rockville, MD, USA).** *p* < 0.01.

**Figure 6 toxins-14-00867-f006:**
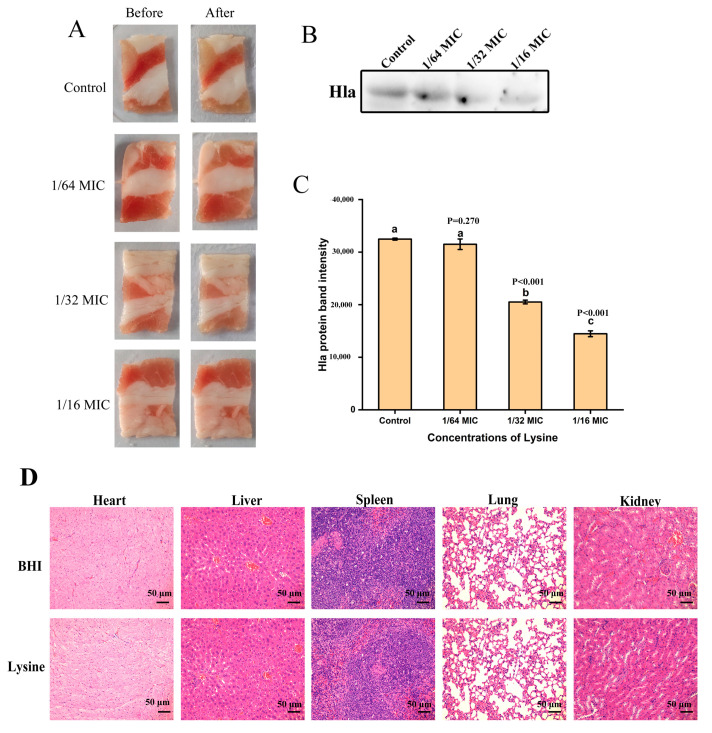
Lys has inhibitory effect on Hla expression of *S. aureus*-contaminated meat model and good biocompatibility. (**A**) Effect of Lys treatment on the appearance of food models. (**B**,**C**) Western blot analysis of the effect of Lys treatment on Hla expression in a food model contaminated with *S. aureus*. There were significant differences between the mean values of unshared letters. (**D**) 5 M/kg of Lys was administered to mice by gavage. After continuous gavage for 30 d, the heart, liver, spleen, lung and kidney of mice were collected and stained with H&E. The scale bar was 50 μm.

**Table 1 toxins-14-00867-t001:** MIC and MBC of Lys on *S. aureus*.

Strain	MIC (mM)	MBC (mM)	Original Source of Strain
ATCC29213	400	400	American Type Culture Collection
265	400	400	Patient vomit
265Δ*sea*	400	400	Constructed and preserved in this laboratory
265Δ*hla*	400	400	Constructed and preserved in this laboratory

Note: 265Δ*sea* was a strain with the *sea* knocked out, 265Δ*hla* was a strain with the *hla* knocked out.

**Table 2 toxins-14-00867-t002:** Sensory evaluation of mutton treated with Lys.

Sample	Average Sensory Score (Points)				
	Color	Organization Status	Odor	Overall Acceptabilit	Total
A	8.214 ± 1.075	7.857 ± 1.424	7.810 ± 1.030	8.024 ± 1.270	8.092 ± 0.258
B	7.762 ± 1.758	7.548 ± 1.870	8.024 ± 1.346	7.595 ± 1.338	7.810 ± 0.188
C	8.405 ± 1.114	8.071 ± 1.132	8.429 ± 1.087	8.190 ± 1.066	8.089 ± 0.222
D	7.929 ± 1.734	7.762 ± 1.602	8.095 ± 1.633	7.619 ± 1.387	7.857 ± 0.257

Note: A is PBS treatment group, B is 1/16 MIC Lys treatment group, C is 1/32 MIC Lys treatment group, D is 1/64 MIC Lys treatment group. (*p* > 0.05)

## Data Availability

Not applicable.

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
