# Peer review of "Lysine Inhibits Hemolytic Activity of *Staphylococcus aureus* and Its Application in Food Model Contaminated with *Staphylococcus aureus"

_toxins, 2022, doi:10.3390/toxins14120867_

Round 1
Reviewer 1 Report
The article explored the potential of lysine to inhibit hemolytic activity of Staph aureus alpha-hemolysin at sub-inhibitory concentrations which do not effect growth of S aureus. Interestingly, the authors found that 1/16 - 1/64 MIC concentrations hemolysis was suppressed. Then authors explored the mechanism behind this, which was shown to inhibit Hla oligomerization. Furthermore, the authors showed this activity of Hla hemoysis suppression in-vitro, and also demonstrated a protective effect of lysine in a mouse model, using supernatants.
Overall, the manuscript is well-written and the experimental approach is sound.
There are a few grammatical issues in the introduction and methods that should be reviewed more closely. Of note, the word "activated" is used, but really should probably be changed to "incubated". Additionally, in line 65 it seems like the word "McFarland" is omitted, as 0.5 McFarland seems like the appropriate term within this sentence.
The authors report on decreases in intra-cellular protein expression by Coomassie staining, it would be nice if the authors provide possible potential mechanisms for this (despite not being explored). Additionally, with the decrease in intra-cellular protein expression and decrease in Hla expression, it is possible that other extra-cellular proteins are inhibited which may include other toxins. This could also explain the in-vivo effects of decreased damage in the ileum and jejunum. This should be mentioned in the discussion section.
Author Response
Response to Reviewer 1 Comments
Dear Reviewers and Editors,
We are very grateful to the reviewers and editors of the paper for their critical reading of the manuscript and many valuable recommendations for our further improvements. We have revised the manuscript and marked changes in blue in revised manuscript according to the comments.
Point 1: Of note, the word "activated" is used, but really should probably be changed to "incubated".
Response 1:Thank you for pointing out this problem. We have revised "activated" to "incubated".
Point 2: Additionally, in line 65 it seems like the word "McFarland" is omitted, as 0.5 McFarland seems like the appropriate term within this sentence.
Response 2:Thank you for pointing out the inappropriateness of our language. We have added "McFarland" to the sentence.
Point 3: The authors report on decreases in intra-cellular protein expression by Coomassie staining, it would be nice if the authors provide possible potential mechanisms for this (despite not being explored). Additionally, with the decrease in intra-cellular protein expression and decrease in Hla expression, it is possible that other extra-cellular proteins are inhibited which may include other toxins. This could also explain the in-vivo effects of decreased damage in the ileum and jejunum. This should be mentioned in the discussion section.
Response 3:Thank you for your suggestion. In the Coomasil bright blue staining results, we found that lysine inhibited the expression of protein in S. aureus cells. Hla, as an important virulence factor of S. aureus, is also contained in intracellular proteins, so lysine inhibits the expression of intracellular proteins, which in turn inhibits the expression of Hla. However, we need to carry out further research on the specific inhibition process and mechanism. In addition to Hla, the toxicity of other toxins may also be inhibited, which may be another possible factor that lysine alleviates the damage of Staphylococcus aureus supernatant to jejunum and ileum. This explanation has been added to the discussion.

Reviewer 2 Report
The manuscript titled “Lysine inhibits hemolytic activity of Staphylococcus aureus and its application in food model contaminated with Staphylococcus aureus” has been reviewed for consideration in toxins. The methodology followed is appropriate, the results are reproduceable. The topic is of the interest in scientists, researchers working in the field of food safety, food toxicity. However, following points should be incorporate before it could be considered again for consideration.
Introduction
Line 18: don’t start sentence with abbreviation.
Line 50-57: The concluding paragraph should be composed with punch line and contained objectives of the study, please rewrite them
Line 66: please write complete name of CFU first place then use symbols
Line 212-214: In statistical analysis ANOVA and t-test were given, please completely describe other important methods of statistics, like how significant difference should be evaluated with groups?
Conclusion: please include conclusion section in the end
References: the reference are not according to journal format (toxins), please check
Author Response
Response to Reviewer 2 Comments
Dear Reviewers and Editors,
We are very grateful to the reviewers and editors of the paper for their critical reading of the manuscript and many valuable recommendations for our further improvements. We have revised the manuscript and marked changes in blue in revised manuscript according to the comments.
Point 1: Line 18: don’t start sentence with abbreviation.
Response 1: Thank you for pointing out this problem. We have changed the abbreviation to the full name.
Point 2: Line 50-57: The concluding paragraph should be composed with punch line and contained objectives of the study, please rewrite them.
Response 2: Thank you for pointing out this problem, we have revised this sentence to "Therefore, the purpose of this study was to evaluate the antibacterial activity of mono-lysine against Staphylococcus aureus, and to further explore the possible mechanisms, using Hla as the entry point".
Point 3: Line 66: please write complete name of CFU first place then use symbols.
Response 3: Thank you for your advice. We have completed the CFU supplement to "colony forming unit (CFU) per mL".
Point 4: Line 212-214: In statistical analysis ANOVA and t-test were given, please completely describe other important methods of statistics, like how significant difference should be evaluated with groups?
Response 4: Thank you for pointing out the inappropriateness of our language. We have completed the detailed analysis. In particular, "For two-group comparison,p values were derived from the one-way Student t test to de-termine differences between groups with normally distributed data. Data with normal distribution were analysed by one-way ANOVA with Dunnett's post-test or Tukey's cor-rection for multiple comparisons".
Point 5: Conclusion: please include conclusion section in the end.
Response 5: Thanks for your suggestion, we have added the conclusion to the manuscript. The conclusion is as follows: As a virulence factor of S. aureus, Hla plays an important role in various infections of S. aureus. In this study, we evaluated the effect of lysine, a nutrient, on the hemolytic activity of S. aureus and explored the mechanisms involved. Our results indicated that lysine inhibited the hemolytic activity of S. aureus by inhibiting the expression of Hla and interfering with the oligomerization of Hla. This allows lysine to alleviate the damage of S. aureus supernatant to mice and Caco-2 cells. In food model applications, lysine exhibited the ability to inhibit Hla expression. At the same time, in animal toxicity experiments, lysine has no obvious toxic effect on the main tissues and organs of mice, indicating that lysine has good biocompatibility and has the potential to be used as an anti- S. aureus preparation in the food industry.
Point 6: References: the reference are not according to journal format (toxins), please check.
Response 6: Thank you for pointing this out. We have looked carefully at the format of the reference and changed it again with the requirements of the "toxins" journal.

Reviewer 3 Report
The reviewer would like to thank the authors for their work entitled, “Lysine inhibits hemolytic activity of Stapholococcus aureus and its application in food model contaminated with Staphylococcus aureus”. The authors hypothesize that monomeric lysine inhibits the activity of alpha hemolysin greater than a media control and that it can increase the survival of mice and help with the preservation of food. The reviewer will provide a critique in both the Style and the Technical aspects of the article.
Style
1. The alt-text for the figures appears to be written in a different language. The review just wanted to point this out in case the authors wish to revise.
2. Generally speaking, for scientific literature it is not really appropriate to start a sentence with a numerical value (ex. Line 102 “10 mL of 256Δsea bacterial suspension….”
It would be better to say, “A 10 mL suspension of 256Δsea bacteria…”
This reviewer would like to politely ask the authors to please revise spots where this occurs.
3. The authors describe at times the concentrations in terms of mM and μg/mL. The reviewer would suggest that they may wish to include in charts where MIC values appear both the mM and μg/mL for the convenience of the readers or at least mention it in the captions.
4. Figure 3C seems to be a little bit hard to read; the reviewer would suggest including a higher-resolution image.
5. In Line 163 the authors mention that “time of death was recorded regularly by a special person” Could they please clarify what is meant by “special person”, the phrasing seems odd.
6. The authors may wish to consider breaking up some of their figures into smaller ones as they contain a lot of information in each.
Technical
1. Caco-2 cells are notorious for their heterogeneous population of cell morphologies and sub-types. How did the authors ensure that their cultures were consistent between experiments? Could they please provide, for instance, the passage number used in each experiment?
2. The reviewer has two questions with respect to the cited reference for the intestinal protection experiments (2.8). Firstly the experimental reference seems to have been performed with S. enterica whereas the present experiments are conducted with S. aureus. Secondly, the reference states that organs were harvested after 48 hrs as opposed to 24 hrs as the authors are doing. Could the authors please comment on these differences and why 24 hrs were used instead of 48 hrs?
3. Examining Figure 1E, it appears that treatment with lysine decreases the overall protein expression, not just Hla. All of the bands appear to be lighter in 1/16 MIC vs 1/32 MIC 1/64 MIC and Could the authors please comment?
4. The crux of the authors argument, that monomeric lysine inhibits Hla in vivo, lies with figure 4B. Unfortunately, the bands with 1/64 MIC and 1/32 MIC seem to have a black dot in them. Could the authors comment on what those dots are, are they actually Hla in their estimation? Would the results of 4C change if those dots were excluded from the calculation of intensity?
5. Line 304, when the authors write, “significantly reduced the expression of Hla in meat) could they please include the p-values?
6. It may also be appropriate to include values of statistical significance (or some sort of indicator such as **) in Figures 1D, 1G, 2B, 2D, 2F, 3C, 4C, and Table 2. The reviewer is confident that the authors have these date; it would be good for the readers to be able to have it on the figures.
7. The authors mention using a 6% Sheep Red Blood Cell (RBC) solution. Could they kindly please clarify approximately home many RBC/mL this would be?
Author Response
Response to Reviewer 3 Comments
Dear Reviewers and Editors,
We are very grateful to the reviewers and editors of the paper for their critical reading of the manuscript and many valuable recommendations for our further improvements. We have revised the manuscript and marked changes in blue in revised manuscript according to the comments.
Style
Point 1: The alt-text for the figures appears to be written in a different language. The review just wanted to point this out in case the authors wish to revise.
Response 1: Thanks for pointing this out. Because the images in the manuscript were generated by different software (Origin and Adobe Photoshop), both with default font output, there may be differences. We will pay attention to the consistency of the fonts in the pictures in the future research.
Point 2: Generally speaking, for scientific literature it is not really appropriate to start a sentence with a numerical value (ex. Line 102 “10 mL of 256Δsea bacterial suspension….”
It would be better to say, “A 10 mL suspension of 256Δsea bacteria…”
This reviewer would like to politely ask the authors to please revise spots where this occurs.
Response 2: Thanks for pointing this out. We have carefully examined the manuscript and revised the parts of the manuscript that have such problems.
Point 3: The authors describe at times the concentrations in terms of mM and μg/mL. The reviewer would suggest that they may wish to include in charts where MIC values appear both the mM and μg/mL for the convenience of the readers or at least mention it in the captions.
Response 3: Thank you for pointing out this problem. We have added the corresponding relationship between mM and MIC in the corresponding places.
Note: 1600 mM is 4 MIC,800 mM is 2 MIC,400 mM is MIC,200 mM is 1/2 MIC,100 mM is 1/4 MIC.
Point 4: Figure 3C seems to be a little bit hard to read; the reviewer would suggest including a higher-resolution image.
Response 4: Thank you for pointing this out. We have broken Figure 3 down into smaller pieces for better interpretation.
Point 5: In Line 163 the authors mention that “time of death was recorded regularly by a special person”. Could they please clarify what is meant by “special person”, the phrasing seems odd.
Response 5: Thank you for pointing this out. We did not express this sentence clearly. What we're talking about is that at certain points in time, professional experimenters record the deaths of mice. We have corrected this part to "and the number of deaths and the time of death were recorded regularly by professional experimenters".
Point 6: The authors may wish to consider breaking up some of their figures into smaller ones as they contain a lot of information in each.
Response 6: Thank you for your advice. We have broken Figure 3 down into smaller pieces for better interpretation.
Technical
Point 1: Caco-2 cells are notorious for their heterogeneous population of cell morphologies and sub-types. How did the authors ensure that their cultures were consistent between experiments? Could they please provide, for instance, the passage number used in each experiment?
Response 1: Thank you for your questions. In order to ensure the consistency of Caco-2 cells, after the cells were resuscitated, we passed them for 3 times to make them stable. The cell morphology was observed under the microscope, and it was confirmed that the cells were in good condition and the morphology was consistent.
Point 2: The reviewer has two questions with respect to the cited reference for the intestinal protection experiments (2.8). Firstly the experimental reference seems to have been performed with S. enterica whereas the present experiments are conducted with S. aureus. Secondly, the reference states that organs were harvested after 48 hrs as opposed to 24 hrs as the authors are doing. Could the authors please comment on these differences and why 24 hrs were used instead of 48 hrs?
Response 2: Thank you for pointing this out. We just took the research idea of this reference for reference, and then, combined with the experience of our research group in the study of Staphylococcus aureus, finally chose 24 h as the experiment time.
Point 3: Examining Figure 1E, it appears that treatment with lysine decreases the overall protein expression, not just Hla. All of the bands appear to be lighter in 1/16 MIC vs 1/32 MIC 1/64 MIC and Could the authors please comment?
Response 3: Thank you for your suggestion. In the Coomasil bright blue staining results, we found that lysine inhibited the expression of protein in S. aureus cells. Hla, as an important virulence factor of S. aureus, is also contained in intracellular proteins, so lysine inhibits the expression of intracellular proteins, which in turn inhibits the expression of Hla. However, we need to carry out further research on the specific inhibition process and mechanism. In addition to Hla, the toxicity of other toxins may also be inhibited, which may be another possible factor that lysine alleviates the damage of Staphylococcus aureus supernatant to jejunum and ileum. This explanation has been added to the discussion.
Point 4: The crux of the authors argument, that monomeric lysine inhibits Hla in vivo, lies with figure 4B. Unfortunately, the bands with 1/64 MIC and 1/32 MIC seem to have a black dot in them. Could the authors comment on what those dots are, are they actually Hla in their estimation? Would the results of 4C change if those dots were excluded from the calculation of intensity?
Response 4: Thank you for your question. This part of the experiment is to detect the expression of Hla in the mutton samples. There are more fats in the mutton samples, which may lead to the uneven distribution of proteins in the samples, and then show the uneven coloring of proteins on the PVDF membrane. We took that into account when we did the grayscale analysis. But if it's excluded, it doesn't affect the overall results.
Point 5: Line 304, when the authors write, “significantly reduced the expression of Hla in meat) could they please include the p-values?
Response 5: Thank you for pointing this out. We have added p-values to the graph and the result.
Point 6: It may also be appropriate to include values of statistical significance (or some sort of indicator such as **) in Figures 1D, 1G, 2B, 2D, 2F, 3C, 4C, and Table 2. The reviewer is confident that the authors have these date; it would be good for the readers to be able to have it on the figures.
Response 6: Thank you for your questions. Corresponding statistically significant values (p-values) have been added to the chart.
Point 7: The authors mention using a 6% Sheep Red Blood Cell (RBC) solution. Could they kindly please clarify approximately home many RBC/mL this would be?
Response 7: Thank you for your questions. The 6% Sheep Red Blood Cell (RBC) solution is approximately 4*108/mL. We've added it to the manuscript.

Round 2
Reviewer 3 Report
The reviewer would like to thank the authors for their responses to questions and still has one remaining question.
The difference between harvesting organs at 24 hrs vs 48 hrs may not be trivial. It would be helpful for the readers if the authors could please provide a reference showing that 24 hrs for harvesting would suffice in this type of model.
Author Response
Dear Reviewers and Editors,
We are very grateful to the reviewers and editors of the paper for their critical reading of the manuscript and many valuable recommendations for our further improvements. We have revised the manuscript and marked changes in blue in revised manuscript according to the comments.
Point 1: The reviewer would like to thank the authors for their responses to questions and still has one remaining question.
The difference between harvesting organs at 24 hrs vs 48 hrs may not be trivial. It would be helpful for the readers if the authors could please provide a reference showing that 24 hrs for harvesting would suffice in this type of model.
Response 1: Thanks again for your advice. First of all, in vitro experiments, we tested the effect of lysine on hemolysin activity after treating S. aureus with lysine for 24 h, so in vivo time was also selected for 24 h. Secondly, studies have shown that the subinhibitory concentration of magnolol can reduce the secretion of S. aureus Hla to weaken the toxicity of S. aureus. The animal experiment in this study was also conducted for 24 h [1]. In addition, in the study on the protection of baicalin against S. aureus pneumonia in mice by inhibiting the cell lysis activity of α-hemolysin, the protective effect of baicalin treatment for 24 h on lung was also studied [2]. At present, there are many studies on natural substances inhibiting S. aureus Hla activity and then inhibiting its pulmonary infection, but there are few articles on natural substances inhibiting S. aureus Hla activity and then inhibiting its intestinal damage. In some studies,in order to evaluate the therapeutic effect of ganoderma lucidum fungal mycelium (MAK) on indomethacin induced ileotis in mice, intestinal inflammation was evaluated in C57BL/6 mice after the macrophages were stimulated by MAK and transferred to them for 24 h [3]. In addition, in order to evaluate the side effects of pemetrexed, fecal flora composition and the integrity of the epithelial barrier in mice after 24 h of treatment were evaluated [4]. The selection of time points for this part of the experiment was also based on the previous experimental ideas and combined with the data of the in vitro experiment. Finally, we modified the expression of this part and added the above references to the manuscript, hoping to provide more references for readers.
- Guo, N.; Liu, Z.; Yan, Z.; Liu, Z.; Hao, K.; Liu, C.; Wang, J. Subinhibitory concentrations of Honokiol reduce alpha-Hemolysin (Hla) secretion by Staphylococcus aureus and the Hla-induced inflammatory response by inactivating the NLRP3 inflammasome. Emerging microbes & infections 2019, 8, 707-716, doi:10.1080/22221751.2019.1617643.
- Qiu, J.; Niu, X.; Dong, J.; Wang, D.; Wang, J.; Li, H.; Luo, M.; Li, S.; Feng, H.; Deng, X. Baicalin protects mice from Staphylococcus aureus pneumonia via inhibition of the cytolytic activity of alpha-hemolysin. The Journal of infectious diseases 2012, 206, 292-301, doi:10.1093/infdis/jis336.
- Nagai, K.; Ueno, Y.; Tanaka, S.; Hayashi, R.; Shinagawa, K.; Chayama, K. Polysaccharides derived from Ganoderma lucidum fungus mycelia ameliorate indomethacin-induced small intestinal injury via induction of GM-CSF from macrophages. Cellular immunology 2017, 320, 20-28, doi:10.1016/j.cellimm.2017.08.001.
- Pensec, C.; Gillaizeau, F.; Guenot, D.; Bessard, A.; Carton, T.; Leuillet, S.; Campone, M.; Neunlist, M.; Blottiere, H.M.; Le Vacon, F. Impact of pemetrexed chemotherapy on the gut microbiota and intestinal inflammation of patient-lung-derived tumor xenograft (PDX) mouse models. Scientific reports 2020, 10, 9094, doi:10.1038/s41598-020-65792-6.
